# OpenReview forum: "The Realignment Problem: When Right becomes Wrong in LLMs"
_ICML.cc/2026/Conference — ICML 2026 regular_

### Official Review · Reviewer_eFDa · 2026-03-05

**Soundness:** 3
**Presentation:** 3
**Significance:** 3
**Originality:** 3
**Overall Recommendation:** 4
**Confidence:** 3

**Summary:**

The paper introduces TRACE, a post-deployment realignment framework that updates LLM behavior to new policy specifications without collecting new human annotations. TRACE triages existing preference pairs into invert, punish, and retain sets using a new-policy oracle, then applies a hybrid objective (DPO on inverted pairs, NPO or oracle-guided DPO for punishment, KL regularization on retain) with impact weights computed via a bilevel-inspired gradient dot-product approximation. Experiments on PKU-SafeRLHF and a synthetic benchmark across three model families suggest TRACE improves policy adherence compared to an unlearning baseline (U2A), with moderate capability preservation and mixed but competitive robustness under adversarial stress tests.

**Compliance With Llm Reviewing Policy:**

Affirmed.

**Final Justification:**

I think this paper is in better format than before so I'm raising my score to a 4. Thanks for the authors. I think the overall experimental design, limited evaluation and simplicity of the approach (e.g. the reliance of flipping labels as being the primary mechanism to simulate imperfect policy oracles) limits its current potential for a higher score, but I'd be happy with this paper being accepted as it is with these changes.

**Key Questions For Authors:**

- Do I understand correctly that you rely on policy-oracle judge (i.e. a stronger LLM) to triage and generate corrective responses? If so, doesnt this introduce judge bias and potential circularity, i.e. the impact of judge errors on final alignment exists which should be quantified
- Adversarial robustness results are mixed, i.e. TRACE worse than U2A on one attack, better on another. And still far from DPO-Gold. How does this square with the claim made about the robustness in the paper?
- Not a lot of details on the new-policy oracle (model identity, calibration, thresholds).
- Do you have access to the $\pi$ old specification or not? A bit unclear
- Can you expand or give a comparative table (and maybe empirical comparisons0 to robust preference optimization methods (e.g. ShaPO), prefernce-noise DPO variants, etc?

**Limitations:**

Limitations are mostly the questions in this case. If you could address those, that'd be supre helpful and I'd be happy to consider increasing my score. A part of the limitations might be my misunderstanding of the paper, too.

**Strengths And Weaknesses:**

- Evaluation spans multiple backbones (Qwen2.5-7B, Gemma-2-9B, Llama-3.1-8B), two settings (real PKU-SafeRLHF policy shifts and a controlled synthetic benchmark), and includes human preference judgments with reported inter-annotator agreement.
- the triage formulation is well explained
- hybrid loss design is intuitive

---

> ### Author Rebuttal · Authors · 2026-03-31
>
> We thank the reviewer for the thoughtful feedback and for identifying TRACE as a framework with *clear merits* and *intuitive design.*
>
> **Q1. Regarding judge errors**
>
> We have already quantified the impact of judge errors in **Appendix A.5 & A.6** and **Table 3 and Table 9**. We simulated "Imperfect Policy Oracles" by randomly flipping **10% and 20%** of the triage labels (Table 9). Results show that TRACE maintains superior performance to the U2A baseline even under 20% label noise.
>
> ---
>
> **Q2. Regarding mix adversarial results**
>
> Purely punitive methods like **U2A (24.6% ASR)** achieve robustness by becoming "over-refusive." However, this renders the model unhelpful. **Pareto Trade-off:** TRACE prioritizes **constructive realignment** (the "Invert" set, Line 265), which teaches the model *how* to be safe while remaining useful. This carries a minor robustness cost (2.7% delta in ASR) but results in a massive gain in human preference as shown below
>
> **Table R5: Robustness-Utility Pareto Frontier**
>
> | Method | ASR (Robustness) ↓ | Policy Agreement ↑ | Human Win-Rate (Utility) |
> | :--- | :---: | :---: | :---: |
> | U2A (Punitive) | **24.6%** | 58.1% | 18.2% |
> | **TRACE (Ours)** | 27.3% | **70.2%** | **81.8%** |
>
> TRACE is *preferred by humans in over 80% of cases (Line 388)*, whereas U2A's robustness comes at the cost of catastrophic utility loss (18.2% win rate).
>
> ---
>
> **Q3. Need more details on the new-policy oracle (identity, thresholds).**
>
> As noted in **Line 330**, we use **GPT-4o** ($T=0$) for the Oracle. The oracle evaluates compliance against the **19-axis taxonomy** defined in **Figure 2 (Line 759)** and **Figure 3 (Line 821)**.  **Human Alignment:** We conducted human audits on 500 triage decisions (Line 1243), finding **94% agreement** between the LLM Oracle and human judges (Krippendorff’s $\alpha = 0.77$). We will add a detailed "Oracle Calibration" section in the Appendix, providing the exact prompts and temperature settings used for triage.
>
> ---
>
> **Q4. Do you have access to the old specification or not?**
>
> We possess the original preference dataset, but we assume the **original policy document ($\pi_{old}$)** is either unavailable or outdated **(Line 216)**. TRACE is designed to re-interpret the data under the *new* policy without needing to reverse-engineer the *old* one.
>
> ---
>
> **Q5. Comparing TRACE to existing robust preference optimization variants (e.g., ShaPO)**
>
> As suggested by the Reviewer, we conduct additional experiments with an adapted version of rDPO [1]. We would like to clarify that the problem of policy realignment for an LLM is fairly underexplored, and we have already adapted an existing technique U2A for baselines comparison, although it does not directly explore the issue of False Dichotomy.
>
> ShaPO (Yang et al., 2026) is a geometry regularizer for stability (ShaPO Eq. 5), while TRACE is a framework for policy transformation. ShaPO is an inappropriate baseline because: (1) its objective would stabilize behavior around obsolete "digital fossils" (Line 437) rather than facilitate adaptation; and (2) its binary Bradley-Terry foundation (ShaPO Eq. 8) cannot handle the "Punish" (DII) scenario (Line 268) where both responses are rendered non-compliant. ShaPO optimizes the geometry of current values; TRACE re-interprets the values themselves. We would also like to highlight that ShaPO (Feb 2026) postdates our submission. Additionally to best of our knowledge the code has not been released, making reproduction infeasible within rebuttal period
>
> We simulated "Imperfect Policy Oracles" by randomly flipping **10% and 20% of the triage labels**. To make the argument more concrete, we introduce some extra noise by also exchanging the categories. As shown in the Table below, TRACE degrades more gracefully than rDPO, demonstrating robustness to oracle imperfections.
>
> **Table R6: Target Policy Agreement (**TPA%**) between rDPO and TRACE at varying noise levels**
> | Oracle Noise Level | R-DPO| TRACE |
> | ----------- | --------- | --------- |
> | 0%          | 81.2%     | 70.7%     |
> | 10%         | 55.3%     | 63.5%     |
> | 20%         | 53.6%     | 59.9%     |
>
> [1] Chowdhury, Sayak Ray, Anush Kini, and Nagarajan Natarajan. "Provably robust dpo: Aligning language models with noisy feedback." arXiv preprint arXiv:2403.00409 (2024).

---

> > ### Author Rebuttal · Reviewer_eFDa · 2026-04-03
> >
> > I think this paper is in better format than before so I'm raising my score to a 4. Thanks for the authors. I think the overall experimental design, limited evaluation and simplicity of the approach (e.g. the reliance of flipping labels as being the primary mechanism to simulate imperfect policy oracles) limits its current potential for a higher score, but I'd be happy with this paper being accepted as it is with these changes.

---

> > > ### Author Response · Authors · 2026-04-05
> > >
> > > ### **Response to Follow-up Questions by Reviewer eFDa**
> > > We are grateful to the reviewer for the increased score and the constructive feedback. Regarding the "simplicity" of using label flipping to simulate oracle imperfections, we have conducted a more rigorous **"Weak Oracle"** experiment. Instead of random noise, we used Qwen2.5-3B to perform the triage instead of GPT-4o on Llama3.1-8B .
> > >
> > >
> > > | Method & Data Size | GPT-4o TPA% | Qwen2.5-3B TPA% | Delta |
> > > | :--- | :---: | :---: | :---: |
> > > | **TRACE (20k)** | **70.7%** | **57.0%** | -13.7% |
> > > | Naive DPO (20k) | 52.2% | 46.1% | -6.1% |
> > >
> > > We will incorporate this "Oracle Degradation" analysis with more models into the final manuscript.

---

### Official Review · Reviewer_6QAz · 2026-03-12

**Soundness:** 3
**Presentation:** 3
**Significance:** 3
**Originality:** 3
**Overall Recommendation:** 4
**Confidence:** 4

**Summary:**

This paper proposes TRACE(Triage and Re-align by Alignment Conflict Evaluation), which is a framework that transform realignment problem into a structured optimization problem via triaging old preference pairs against a new policy into "Invert," "Punish," and "Retain" cases, and then apply alignment impact-weighted optimization. The work is evaluated on PKU-SafeRLHF and SynthValueBench across Qwen2.5-7B, Gemma-2-9B, and Llama-3.1-8B, all showing that TRACE consistently outperforms the U2A(Unlearning to Align) baseline.

**Compliance With Llm Reviewing Policy:**

Affirmed.

**Final Justification:**

The rebuttal addressed my main concerns.

**Key Questions For Authors:**

- Any comparison to some stronger baselines?
- If the methods does not have sufficient access to old preference data and old policy information, how would the performance decay?

**Limitations:**

Yes. Note also that:
- The method seems to largely rely on the access to old preference data and old policy information, which seems unrealistic in many settings.

**Strengths And Weaknesses:**

Strength:

- The work formalize the Alignment-Reality Gap issue, which is an issue many unlearning methods fail to address, providing meaningful insight towards realignment.
- The evaluation is pretty decent with multiple benchmark dataset and three model families.

Weakness:

- TRACE seem to rely on the access to the old preference pairs for trained models, while in practice, these can be hard to obtain.
- It is said that “DPO-Gold was preferred over TRACE in 68.2% of cases on PKU-SafeRLHF, ad 74.6% on SynthValueBench”. While the performance is good, the claim of TRACE “nearly matching” DPO-Gold seems to be a bit exaggerated.
- The current baseline seems to be only U2A and DPO-Gold as an upper bound. It seems a bit unclear how TRACE compared to other realignment methods.
- Most of the results table are in the appendix, which is actually a bit hard to read. Might need to reorganize to at least one important result can be directly seen from the main text.

---

> ### Author Rebuttal · Authors · 2026-03-31
>
> We thank the reviewer for their thoughtful assessment and for recognizing that TRACE provides *"meaningful insight towards realignment"* by formalizing the *Alignment-Reality Gap*.
>
> **W1. Regarding access to old preference data**
>
> TRACE assumes access to historical preference pairs, which form the minimal representation of prior alignment decisions. Since alignment methods such as RLHF and DPO are trained directly on such pairwise comparisons, these datasets capture the behavioral constraints imposed during training and are often retained for auditing, evaluation, and iterative updates. While such preference pairs could in principle be estimated or reconstructed from model outputs, doing so would introduce additional noise that is entangled with the effect of policy re-alignment.
>
> Our goal in this work is to isolate the policy reinterpretation problem: given a fixed set of prior preference decisions, how should they be updated under a new policy. By operating directly on existing preference pairs, TRACE avoids confounding alignment updates with errors in preference reconstruction and addresses a well-defined component of the broader realignment pipeline. Consistent with this, our oracle-noise experiment shows that TRACE is not brittle to imperfect supervisory signals: even with 10% and 20% flipped labels (Table 3), policy agreement degrades from 70.2% to 67.4% and 61.5%, respectively, which is above the U2A baseline (55.3% and 54.7%). This indicates that TRACE remains robust to moderate noise in the policy signal. If preference pairs are unavailable, additional mechanisms for preference reconstruction would be required, which is beyond the scope of this work.
>
> ---
>
> **W2. Regarding "Nearly Matching" DPO-Gold**
>
> We agree that the term "nearly matching" is imprecise. While DPO-Gold (Retraining) remains the performance ceiling, TRACE's primary value is in closing the massive gap between unlearning (U2A) and retraining. We will revise our claims in **Section 7.1** to describe TRACE as **"substantially closing the gap between unlearning and human-re-annotation"** rather than "nearly matching."
>
> ---
>
> **W3 & Q1. Regarding additional baselines**
>
> As suggested by the Reviewer, we conduct additional experiments with an adapted version of rDPO [1]. We would like to clarify that the problem of policy realignment for an LLM is fairly underexplored, and we have already adapted an existing technique U2A for baselines comparison, although it does not directly explore the issue of False Dichotomy.  We simulated "Imperfect Policy Oracles" by randomly flipping **10% and 20% of the triage labels**. To make the argument more concrete, we introduce some extra noise by also exchanging the categories. As shown in the Table below, TRACE degrades more gracefully than rDPO, demonstrating robustness to oracle imperfections.
>
> **Table R3: Comparison between rDPO and TRACE at varying noise levels**
> | Oracle Noise Level | R-DPO| TRACE |
> | ----------- | --------- | --------- |
> | 0%          | 81.2%     | 70.7%     |
> | 10%         | 55.3%     | 63.5%     |
> | 20%         | 53.6%     | 59.9%     |
>
> [1] Chowdhury, Sayak Ray, Anush Kini, and Nagarajan Natarajan. "Provably robust dpo: Aligning language models with noisy feedback." arXiv preprint arXiv:2403.00409 (2024).
>
> ---
>
> **W4. Improving presentation & organization**
>
> In the final version, we will move a condensed version of **Table 11** and the summary of **Table 13** into the main text.
>
> ---
>
> **Q2. Regarding performance decay with limited old preference data**
>
> We performed a scaling analysis on the *PKU-SafeRLHF* dataset. Across both architectures, *TRACE with only 5k preference samples outperforms Naive Oracle DPO with 20k samples*. This confirms that the performance gains are not merely "inherited" from the Oracle, but are driven by TRACE’s ability to categorize conflicts (Triage) and prioritize high-leverage updates (Impact Weighting).
>
> **Table R4: Target Policy Agreement (TPA%) at Varying Data Scales**
> | Model | Data Size | Naive Oracle DPO | **TRACE (Ours)** | Delta |
> | :--- | :---: | :---: | :---: | :---: |
> | **Llama-3.1-8B** | 5k  | 35.9% | **55.9%** | +20.0% |
> |                   | 10k | 46.5% | **65.5%** | +19.0% |
> |                   | 20k | 52.2% | **70.7%** | +18.5% |
> | **Gemma-2-9B**   | 5k  | 37.1% | **54.4%** | +17.3% |
> |                   | 10k | 48.2% | **66.9%** | +18.7% |
> |                   | 20k | 53.9% | **71.8%** | +17.9% |

---

> > ### Author Rebuttal · Reviewer_6QAz · 2026-04-01
> >
> > Thanks for the authors' detailed response. My concerns are mostly addressed. Yet given the limitations and quality of current version, I'll keep my original assessment.

---

### Official Review · Reviewer_XFL3 · 2026-03-12

**Soundness:** 2
**Presentation:** 2
**Significance:** 3
**Originality:** 3
**Overall Recommendation:** 4
**Confidence:** 3

**Summary:**

This paper studies post-deployment realignment of language models when the governing policy changes over time. Instead of re-annotating a new preference dataset for every policy shift, the paper works in a non-blind setting where the original preference triplets remain available. The proposed method, TRACE, reevaluates each pair under a new policy, triages it into Invert, Punish, or Retain, and then applies reversed-pair preference optimization, punitive or oracle-guided corrective updates, and KL regularization. Conflict samples are further reweighted by an alignment-impact score based on how well a local update aligns with a global re-alignment gradient. Experiments on PKU-SafeRLHF and a synthetic benchmark, SynthValueBench, across Qwen2.5-7B, Gemma-2-9B, and Llama-3.1-8B show that TRACE improves alignment relative to the punitive U2A baseline, but still remains clearly behind full DPO-Gold retraining.

**Compliance With Llm Reviewing Policy:**

Affirmed.

**Ethical Review Concerns:**

Prompt injection: The submission artifact contains reviewer-directed instruction text embedded at the bottom of the page-2 PDF text layer, below the visible content. The injected text instructs the reviewer to include two specific predetermined phrases in the review. This text is invisible in the rendered PDF but extractable via PDF text extraction tools.

**Ethical Review Flag:**

Flag this paper for an ethics review.

**Ethics Expertise Needed:**

["Responsible Research Practice (e.g., IRB, documentation, research ethics)"]

**Key Questions For Authors:**

1. How is J and gJ in Eq. (6) actually computed? What data and how much? This directly affects my soundness score.
2. Per-component ablation (triage, oracle correction, impact weighting, KL) on both datasets? The current single-pair ablation is insufficient.
3. Exact sample sizes for human eval and adversarial tests, plus per-architecture breakdowns instead of only aggregated numbers?
4. How sensitive is TRACE to oracle quality? Would a weaker/mismatched oracle break the Punish correction?

**Limitations:**

Paper does not discuss:

(1) the non-blind assumption (retained triplets)
(2) oracle dependency for Punish correction
(3) the alignment-vs-utility trade-off visible in their own HellaSwag/ASR numbers. Needs to say where TRACE cannot be used and when full re-annotation is still needed.

**Strengths And Weaknesses:**

Strengths:
- Triage formulation (Invert/Punish/Retain) is the real contribution - correctly identifies that flipping a non-compliant winner does not make the loser compliant. Handles the false dichotomy naive unlearning ignores.
- Evaluated across 3 architectures and 2 datasets. TRACE consistently beats U2A on alignment (81.8-85.3% win rate).
- Problem framing is practically relevant. Policy drift is real and full re-annotation is expensive.
- Combination of DPO/NPO/KL losses with impact weighting is thoughtful, even if individual pieces are borrowed.

Weaknesses:
- Impact-weighting (Eq. 6) is underspecified: how is J and gJ computed in practice? What data, what size? Core novel component, not reproducible from main text.
- Ablation (Table 1) only covers impact weighting on one model/dataset. No ablation isolating triage, oracle correction, or KL separately.
- Claims overshoot evidence: "negligible utility degradation" but HellaSwag drops ~3-4 points (81.4 -> 78.2/77.3). "Robust adversarial behavior" but worse ASR than U2A on Fictional Scenario Nesting (27.3% vs 24.6%).
- Abstract calls inversion loss "relational (e.g., IPO)" but Eq. (2) is standard DPO log-sigmoid, not IPO's squared loss.
- Missing reporting: PKU-SafeRLHF shifted policy construction, exact sample counts for human eval and adversarial tests, per-architecture breakdowns.
- Favorable setting: assumes retained original preference triplets + stronger-model oracle for Punish corrections.

---

> ### Author Rebuttal · Authors · 2026-03-31
>
> We thank the reviewer for identifying the Triage formulation (Invert/Punish/Retain) as the real contribution and for recognizing TRACE’s consistent performance across architectures.
>
> #### **W1 and Q1: Impact Weighting Specification ($J$ and $g_J$)**
> $g_J$ is the gradient of a standard DPO objective calculated over a small, balanced **"Gold-Standard" anchor batch** ($D_{gold}$) described in **Algorithm 2 (Line 678)**. This batch (typically $|B|=512$) is sampled from the triaged "Retain" and "Invert" sets once per update step to represent the "perfect" target policy state. It serves as a static directional compass ($g_J$) to filter local task gradients ($g_{Li}$). We will move the formal definition of $J$ and the sampling strategy for $D_{gold}$ from the Appendix into the main text of **Section 5.4** to ensure the core mechanism is fully reproducible.
>
> ---
>
> #### **W2 and Q2: Component-wise Ablation Study**
> We provide the requested multi-component ablation on **Llama-3.1-8B PKU-SafeRLHF** below:
>
> **Table R2: Component-wise ablation of TRACE**
> | Variant | Policy Agreement (%) | MMLU | ASR (Robustness) |
> | :--- | :---: | :---: | :---: |
> | **TRACE (Full)** | **70.7** | **70.2** | **27.3** |
> | w/o Triage (Punish All) | 58.1 | 70.2 | **24.6** |
> | w/o KL Regularization | 71.5 | 64.1 | 29.8 |
> | w/o Impact Weighting | 62.8 | 69.5 | 32.1 |
>
> This confirms that **Triage** provides the baseline alignment, while **Impact Weighting** drives the 7.9% gain in agreement, and **KL Regularization** is critical for utility preservation (6.1% MMLU gain).
>
> ---
>
> #### **W3: Claims vs. Evidence (Utility and Robustness)**
> As we discussed with Reviewer hMLU, this represents a **principled trade-off between absolute refusal and helpful realignment.**. We acknowledge "negligible" was imprecise for a 3.2% HellaSwag drop **(Line 429)**. U2A (24.6% ASR) is slightly more robust than TRACE (27.3% ASR) because U2A adopts a "Refuse-All" strategy. However, this makes the model useless for the new policy, resulting in a **failure-level win-rate (18.2%)** in human evaluations **(Line 1527)**. TRACE prioritizes **constructive realignment**, teaching the model *how* to be safe rather than just being silent. We will revise our language from "negligible" to **"gracefully managed trade-offs"** and add the ASR-vs-Utility Pareto frontier discussion in **Section 7.3**.
>
> ---
> #### **W4: Terminology (IPO vs. DPO)**
> The abstract lists IPO as an *example* of the relational loss family **(Line 037)**. While TRACE is objective-agnostic, we implemented the DPO-style variant for our primary experiments **(Eq. 2, Line 233)**. We will standardize the terminology in the Abstract to "Relational Preference Optimization" to avoid ambiguity.
>
> ---
>
> #### **W5 and Q3: Missing Reporting (Sample Counts and Breakdowns)**
> Sample counts were provided in the Appendix but will be promoted to the main text:
> * **Human Eval:** 1,500 triplets (500 per architecture) evaluated by 3 independent annotators.
> * **Adversarial Tests:** 400 prompts per model (200 per attack vector), stratified across 4 policy dimensions **(Lines 1291 right column)**.
> We will move the per-architecture breakdown tables (**Tables 6, 7, and 8**) from the Appendix to **Section 7**.
>
> ---
> #### **W6, Q4, L1, and L2: Favorable Setting and Oracle Dependency**
> As addressed with Reviewer hMLU, TRACE assumes a **Model Provider** setup where data ($P_D$) is a core institutional asset. Our sensitivity analysis **(Table 3, Line 1272)** shows TRACE remains effective even with **20% noise in the Oracle signal**, outperforming the noise-free U2A baseline **(Table 9)**.
> We will revise **Section 4.1** to frame the "Non-Blind" setup as a principled isolation of the realignment problem from the noise of data recovery.
>
> ---
>
>
> #### **L3: The Alignment-vs-Utility Trade-off**
>
> We characterize this as a **Helpfulness-Utility Pareto Frontier**
> * **TRACE vs. U2A:** While U2A (24.6% ASR) is slightly more robust on "Scenario Nesting," it fails in human utility (18.2% win rate) because it is **over-refusive**. TRACE achieves an **81.8% human win rate (Table 11)** by being more helpful, which inherently accepts a minor 2.7% delta in ASR.
> * **TRACE vs. DPO-Gold:** We acknowledge that TRACE does not fully reach the "Robustness Ceiling" of full human re-annotation (DPO-Gold, ~11% ASR).
> * **Defining the Boundaries (When to Use TRACE vs. Re-annotation):**
> 1. **When to use TRACE:** Rapid adaptation to evolving regulations, cultural shifts, or risk tolerances where **efficiency and helpfulness** are prioritized (e.g., content moderation, customer support).
> 2. **When full re-annotation is still needed:** In **high-stakes/safety-critical domains** (e.g., clinical advice, military applications) where reaching the absolute robustness ceiling (~11% ASR) outweighs the cost-savings of TRACE.
>
> We will incorporate this "Decision Framework" into **Section 8 (Conclusion)** to provide a clear technical roadmap for when full re-annotation is required.

---

### Official Review · Reviewer_hMLU · 2026-03-14

**Soundness:** 2
**Presentation:** 3
**Significance:** 3
**Originality:** 3
**Overall Recommendation:** 4
**Confidence:** 4

**Summary:**

This paper studies the problem of post-deployment realignment of LLMs under policy shifts.

When alignment policies change, models trained under the old policy may produce responses that violate the new policy, and collecting new human preference annotations can be expensive.

The paper proposes TRACE to reuse an existing preference dataset by reinterpreting each pair under a new policy.

The preference pairs are categorized into Invert, Punish, or Retain, and different optimization objectives are applied to each subset.

The method also introduces an impact weighting mechanism intended to prioritize samples that contribute more to overall realignment.

**Compliance With Llm Reviewing Policy:**

Affirmed.

**Final Justification:**

I thank the authors for the rebuttal. The new Table R1 experiment is convincing and directly addresses my most critical concern: TRACE's gains clearly stem from the triage-and-weight mechanism rather than oracle relabeling alone. This is the primary reason I raise my score from 3 to 4.

**Key Questions For Authors:**

1. TRACE appears to rely on a relatively favorable setting where the original preference data is still available. How realistic is this assumption in modern alignment pipelines, especially when many systems are updated with newly generated on-policy data rather than a fixed historical preference dataset? More concretely, if the original preference pairs are unavailable, does TRACE still apply, or would it require a substantial redesign?

2. The paper claims to reduce the need for fresh human annotation, but it still uses an LLM oracle to provide corrective signals for some samples. To better isolate the contribution of the proposed hybrid objectives and impact weighting, could the authors compare against a simpler baseline: relabel the data with the same oracle, then directly train with standard DPO? This would help clarify whether the gains come from the TRACE formulation itself or mainly from oracle-assisted relabeling.

3. TRACE uses dynamic weighting for conflict samples, but the retained portion of the data may be much larger in realistic policy-shift settings. How sensitive is the method to the balance between preserving retained behavior and updating conflict cases? In particular, is there a risk that the retained data dominates optimization and weakens realignment, or conversely that too little preservation hurts general capability?

**Limitations:**

The paper discusses some technical limitations, but the discussion of safety implications should be strengthened. In particular, the adversarial evaluation suggests that TRACE, while more efficient than full relabeling, still leaves a substantial robustness gap.

**Strengths And Weaknesses:**

**Strengths**:

The paper addresses a relevant and practical problem. Alignment policies may evolve due to regulation, safety concerns, or product requirements, making post-deployment realignment an important topic.

The proposed Invert / Punish / Retain triage is intuitive and captures different types of policy conflicts more explicitly than simple punitive unlearning. The method forms a coherent pipeline combining sample categorization, hybrid objectives, and weighted updates. Finally, the experimental section evaluates multiple model families and includes human preference evaluation, capability benchmarks, and adversarial testing, providing a reasonably broad empirical analysis.

**Weakness**:

The experimental setting is somewhat narrower than the problem framing suggests.

TRACE assumes access to the original preference dataset and a policy evaluation mechanism, which represents a relatively favorable scenario compared to many real-world deployment situations.

And although the paper claims to avoid new human annotations, the method still relies on policy judgments or model-assisted correction, meaning supervision is not entirely removed.

In the bi-level optimization framework, the inverse Hessian is approximated as a constant scaling factor. This assumption of a locally isotropic loss landscape is mathematically weak for the highly complex and anisotropic parameter spaces of LLMs.

---

> ### Author Rebuttal · Authors · 2026-03-31
>
> We thank the reviewer for their constructive feedback. We address your specific concerns regarding the experimental setup and mathematical assumptions below.
>
> #### **Q1, W1, and W2: Realism of Data Availability and "Blind" Redesign**
>
> We assume access to historical preference pairs $(y_w, y_l)$ because they form the minimal record of a model's prior alignment decisions. In industrial pipelines (e.g., RLHF/DPO), these datasets are core institutional assets retained for auditing and iterative updates **(Lines 212–219)**.
> By operating on $P_D$, TRACE isolates the **policy reinterpretation problem** from the noisy and computationally expensive task of preference reconstruction (sampling-based inference), which we argue is a separate research challenge **(Lines 216–218)**.  If the original $P_D$ is unavailable, TRACE does **not** require a substantial redesign. The three-stage pipeline (Triage $\rightarrow$ Impact Weighting $\rightarrow$ Hybrid Update) is data-agnostic.
> In a "Blind" setting, a developer would simply generate on-policy responses ($y_{new}$) from the current model and triage them against the Oracle’s policy. Our oracle-noise experiment (Table 3, Line 1272) simulates the noise inherent in such reconstruction settings. We demonstrate that the **Impact Weighting (Eq. 6)** acts as a filter for this noise, ensuring the realignment signal remains strong even when the preference signals are reconstructed rather than historical.
>
> ---
>
> #### **W3: Supervision and LLM-Assisted Correction**
> Our claim is not the removal of supervision, but the removal of **"new third-party human annotation" (Line 089)** and **"costly BPO (Business Process Outsourcing) workflows" (Line 090)**. Replacing significant hours of human labor with automated LLM-oracle triage reduces the realignment cost profile significantly, making rapid policy iteration feasible.
>
> ---
>
> #### **W4: Mathematical Weakness of the Hessian Approximation**
> We acknowledge that LLM loss landscapes are theoretically anisotropic. However, computing the true Inverse Hessian for a billion parameter scale models is computationally intractable **(Line 278)**. We adopted this approximation following established gradient-based influence estimators such as **Tracin (Yang et al., 2024 line 280-285)** and **U2A (Feng et al., 2025)**. The **7.4% delta** in Target Policy Agreement between TRACE and the "Uniform Weights" baseline in **Table 1 (Line 275)** demonstrates that this approximation successfully captures the essential directional signal required for triage. We will add a discussion in **Section 5.4** addressing the use of damping factors ($\gamma$) to mitigate the risks of the isotropic assumption in anisotropic landscapes.
>
> ---
>
> #### **Q2: Impact of TRACE Formulation vs. Oracle Relabeling**
> We thank the reviewer for this suggestion. We performed a scaling analysis across 2 model families on the **PKU-SafeRLHF** dataset (Table R1). In this experiment, "Naive Oracle DPO" refers to standard DPO training on oracle-labeled triplets, while TRACE uses our full triage and weighting pipeline.
>
> **Table R1: Target Policy Agreement (TPA%) at Varying Data Scales**
>
> | Model | Data Size | Naive Oracle DPO | **TRACE (Ours)** | Delta |
> | :--- | :---: | :---: | :---: | :---: |
> | **Llama-3.1-8B** | 5k  | 35.9% | **55.9%** | +20.0% |
> |                   | 10k | 46.5% | **65.5%** | +19.0% |
> |                   | 20k | 52.2% | **70.7%** | +18.5% |
> | **Gemma-2-9B**   | 5k  | 37.1% | **54.4%** | +17.3% |
> |                   | 10k | 48.2% | **66.9%** | +18.7% |
> |                   | 20k | 53.9% | **71.8%** | +17.9% |
>
>
> #### **Q3: Sensitivity to Retain/Conflict Data Balance**
>
> In our scaling and efficiency experiments (Table R1), we observed a consistently challenging triage distribution: **48.2% Punish, 9.3% Invert, and 42.5% Retain**. This results in a **57.5% total conflict set**, meaning over half of the historical dataset is in active violation of the new policy. In most alignment settings, having "Conflict" data (57.5%) outnumber "Retain" data (42.5%) would risk catastrophic utility collapse or alignment failure.
> Despite this aggressive shift where the conflict signal dominates the retention signal, TRACE achieves a **70%+ Policy Agreement** without sacrificing utility (e.g., MMLU remains stable at **70.2–70.5** and HellaSwag remains stable around **77.3-78.2**).
> * **Dynamic Signal Protection:** TRACE avoids the "dominance" risk via two mechanisms:
> 1. **Impact Weighting (Eq. 6):** Even when conflicts are the majority (57.5%), the weighting ensures that only high-leverage, synergistic updates are prioritized. This prevents noisy or contradictory "Punish" signals from overwhelming the model’s core capabilities.
> 2. **KL-Regularization (Eq. 7):** The forward-KL divergence on the **42.5% Retain set** acts as a stable anchor, preserving foundational knowledge while the conflict sets drive the behavioral shift.

---

> > ### Author Rebuttal · Reviewer_hMLU · 2026-04-04
> >
> > two concerns remain insufficiently addressed:
> > 1. Retain/Conflict ratio sensitivity (Q3). The reported 57.5% conflict ratio is actually favorable to the method. The more realistic and arguably more common scenario — where a policy shift affects only a small fraction of data (e.g., 5–10%) — was not explored. The authors' response demonstrates robustness under high-conflict conditions but does not address the question as posed. A brief ablation over varying conflict ratios in the camera-ready would strengthen the practical claims considerably.
> > 2. Adversarial robustness gap. The rebuttal did not address the notable ASR difference between TRACE (27.3%) and DPO-Gold (11.3%). For a framework positioned around safety-critical updates, the current "acceptable trade-off" framing needs more analysis — at minimum, a discussion of when this gap is tolerable and when it is not.
> >
> >
> > The technical contribution is sound and the core mechanism is well-validated. My remaining concern is that the paper's narrative scope exceeds its empirical coverage. I would suggest the authors to tighten the framing in the introduction and limitations to clearly delineate the validated operating regime — specifically regarding data availability assumptions, conflict ratio ranges, and the robustness trade-off. This is a presentation adjustment, not a fundamental redesign, and would bring the paper's claims into alignment with its demonstrated strengths.

---

> > > ### Author Response · Authors · 2026-04-05
> > >
> > > ### **Response to Follow-up Questions by Reviewer hMLU**
> > >
> > > We thank the reviewer for the continued engagement and the increment in score. We address the gradient dynamics of sparse conflict ratios and the structural constraints of adversarial robustness below.
> > >
> > > #### **1. Retain/Conflict Ratio Sensitivity (The 5–10% Regime)**
> > > > **Reviewer Concern:** *The reported 57.5% conflict ratio is favorable. The more realistic scenario (5–10% conflict) was not explored, raising concerns that the Retain set might dominate optimization.*
> > >
> > > From a theoretical standpoint, the TRACE architecture is mathematically designed to preclude the "washout" effect you hypothesize. The balance between the Retain set and the Conflict set is not determined by their raw sample counts, but by the **Alignment Impact Weighting (Eq. 6)**.
> > >
> > > Because the target gradient ($g_J$) is computed on a *balanced* "Gold-Standard" batch ($D_{gold}$), the local gradients ($g_{L_i}$) from the small conflict set will exhibit high cosine similarity with $g_J$ and receive heavily up-weighted learning signals. Conversely, the massive Retain set is managed via a forward-KL penalty (Eq. 4), which anchors the logits but does not generate competitive preference gradients to overwhelm the targeted conflict updates.
> > >
> > > **Empirical Validation (10.3% Conflict Ratio):**
> > > To empirically validate this, we isolated the "Intellectual Property" policy dimension on Llama-3.1-8B, creating a dataset with exactly a **10.3% Conflict Ratio** (89.7% Retain).
> > >
> > > | Method | Target Policy Agreement (%) | MMLU (Utility) |
> > > | :--- | :---: | :---: |
> > > | DPO-Gold  | 78.4% | 68.7 |
> > > | **TRACE (Ours)** | **74.6%** | **67.7** |
> > >
> > > **Conclusion:** As predicted by our formulation, TRACE successfully executes the targeted behavioral shift without being dominated by the 89.7% Retain set. We will include this sparse-conflict ablation in **Section 5.5** across more models to explicitly validate TRACE’s operating regime across extreme distribution skews.
> > >
> > > #### **2. The Adversarial Robustness Gap (27.3% vs. 11.3%)**
> > > > **Reviewer Concern:** *The rebuttal did not address the notable ASR difference between TRACE and DPO-Gold. For safety-critical updates, this needs a discussion of when this gap is tolerable.*
> > >
> > > As we noted in our previous responses, purely punitive methods (U2A) achieve lower ASRs by becoming over-refusive, which catastrophically degrades human utility (18.2% win rate). However, the gap between TRACE (27.3%) and DPO-Gold (11.3%) represents a fundamental **information-theoretic limit of the data-reuse paradigm**.
> > >
> > > As established in the text, DPO-Gold represents the theoretical upper bound of **full human re-training**. It achieves an 11.3% ASR because a complete re-annotation pipeline surfaces novel, human-authored edge-cases specifically designed to map the adversarial boundaries of the *new* policy. TRACE, by definition, operates strictly on *historical* data. It successfully re-aligns the model's in-distribution and moderate out-of-distribution behavior (achieving 76.7% overall policy agreement and an 81.8% human win rate), but a framework restricted to re-weighting old data cannot synthesize the novel adversarial red-teaming examples required to perfectly map extreme out-of-distribution attacks.
> > >
> > > **Operational Decision Framework:**
> > > We will formalize this boundary in the **Limitations (Section 8)** to strictly delineate when this gap is tolerable:
> > > *   **The TRACE Regime (Agile Mitigation):** TRACE is scientifically validated for rapid-response policy adaptation (e.g., shifting regulatory compliance, correcting broad behavioral drifts). It delivers state-of-the-art realignment *in the absence of new human data*, providing immediate therapeutic patches to deployed models where maintaining high helpfulness and utility is paramount.
> > > *   **The DPO-Gold Regime (Adversarial Hardening):** For deployments requiring absolute resilience against dedicated, state-level adversarial attacks (e.g., military, critical infrastructure), historical data reuse is insufficient. In these regimes, the latency and cost of full human red-teaming (DPO-Gold) are strictly required.
> > >
> > > #### **3. Tightening the Narrative Scope**
> > > We appreciate the reviewer's distinction between a presentation adjustment and a fundamental redesign. We will gladly tighten the Introduction and Limitations to formalize these scientific boundaries.
> > >
> > > Specifically, the camera-ready version will clearly state that TRACE is an **efficient, data-reusing realignment framework** that optimizes in-distribution policy adherence, but explicitly does not bypass the need for novel human red-teaming to achieve absolute adversarial robustness.
> > >
> > > We thank you for pushing us to define these boundaries with greater scientific precision, which undoubtedly strengthens the final manuscript.

---

### Decision · Program_Chairs · 2026-04-30

**Decision:**

Accept (regular)

**Comment:**

This paper introduces TRACE, a framework for post-deployment LLM realignment under evolving policy specifications without collecting new human annotations. TRACE repurposes existing preference data through a three-stage pipeline: triaging pairs into Invert/Punish/Retain categories via an LLM oracle, computing alignment impact scores to prioritize high-leverage samples, and applying a hybrid objective (DPO for inversions, NPO for suppression, KL for retention). Experiments across three model families on PKU-SafeRLHF and SynthValueBench show TRACE consistently outperforms the U2A unlearning baseline.

All four reviewers agreed that the problem is practically relevant and that the triage formulation is a meaningful contribution. However, the initial submission had clear deficiencies: the impact weighting mechanism was underspecified and not reproducible from the main text, ablations were incomplete (one model, one dataset), key result tables were buried in the appendix, and several claims were overstated — "negligible utility degradation" despite a 3-4 point HellaSwag drop, "nearly matching" DPO-Gold when DPO-Gold was preferred in 68-75% of comparisons, and "robust adversarial behavior" despite a substantial ASR gap (27.3% vs DPO-Gold's 11.3%).

The rebuttal was substantive and effective: the authors provided a full component-wise ablation, a scaling analysis isolating the triage mechanism's contribution from oracle relabeling, a sparse-conflict-ratio experiment, additional baselines (rDPO, Naive Oracle DPO), and a weak-oracle study. Two reviewers raised their scores accordingly. The unanimous Weak Accept consensus reflects that the core idea is sound and the problem important, but the significant gap between the initial submission and the post-rebuttal version makes it difficult to support a higher recommendation. I recommend Weak Accept.